# Assessment of relationship between retinal perfusion and retina thickness in healthy children and adolescents

**Farhad Salari[1], Vahid Hatami[1], Masoumeh Mohebbi[1,2], Fariba Ghassemi©[1,3]***

**1** Eye Research Center, Farabi Eye Hospital, Tehran University of Medical Sciences, Tehran, Iran, **2** Cornea Service, Farabi Eye Hospital, Tehran University of Medical Sciences, Tehran, Iran, **3** Retina & Vitreous Service, Farabi Eye Hospital, Tehran University of Medical Sciences, Tehran, Iran

* fariba.ghassemi@gmail.com

**Data Availability Statement:** All relevant data are within the manuscript and its Supporting Information files.

## Abstract

### Purpose

To determine the correlations between inner, mid, and outer retinal thickness (RT) and allied retinal and choroidal vascular densities (VD) at macula in normal healthy children and adolescents.

### Methods

This cross-sectional study included a total of 108 eyes of 59 subjects. Optical coherence tomography angiography (OCTA-Optovue) was used to measure the thickness of the inner-retina (IRT), mid-retina (MRT), and outer-retina (ORT) at foveal (central 1mm), parafoveal (1–3 mm), and perifoveal (3–6 mm) areas, as well as the corresponding VD of the superficial capillary plexus (SVD), deep capillary plexus (DVD), and choricapillaris (CVD).

### Results

The study enrolled 108 normal eyes from 54 participants with a mean age of 10.9 years. Partial correlations showed that the nasal and inferior parafoveal and perifoveal subsegments IRT, MRT and ORT are more affected by all SVD, DVD, and CVD. Nasal parafoveal and perifoveal MRT and all three capillary layers have a constant negative correlation. ORT was not affected by all three layers except for CVD at fovea. The regression analysis revealed that SVD and CVD were significantly associated with foveal and parafoveal and perifoveal IRT. DVD and gender could significantly affect perifoveal IRT. However, only CVD was significantly affected foveal MRT. Based on regression analysis, only CVD was significantly associated with foveal and parafoveal ORT, but not with perifoveal ORT.

### Conclusion

The thickness of different retinal layers correlates with retinal and choroidal VD in different ways according to their zones.

**Funding:** The authors received no specific funding for this work.

**Competing interests:** The authors have declared that no competing interests exist.

**Abbreviations:** AMD, Age related macular degeneration; BRM, Bruch's membrane; CC, choriocapillaris; CNV, Choroidal neovascularization; CVD, Choriocapillary vascular density; DCP, Deep capillary plexus; DVD, Deep capillary plexus vascular density; GCC, Ganglion cell complex; GCL, Ganglion cell layer; GEE, Generalized estimating equation analysis; INL, Inner nuclear layer; IPL, Inner plexiform layer; IRT, Inner retinal thickness; MRT, Midretinal thickness; OCTA, Optical coherence tomography angiography; OPL, Outer plexiform layer; ORT, Outer retinal thickness; RPE, Retinal pigmentary epithelium; RT, Retinal thickness; SCP, Superficial capillary plexus; SVD, Superior capillary plexus vascular density; VD, Vascular density; VEGF, Vascular endothelial growth factor.

## Introduction

The retinal vasculature is made up of well-defined capillary networks that deliver oxygen and nutrients to the neuroretina. It is known that vessels from the central retinal artery and cilioretinal artery nourish the macula through two vascular networks. These layers are superficial capillary plexus (SCP- mainly in the ganglion cell layer (GCL)) and deep capillary plexus (DCP) above and below the inner nuclear layer (INL). Also, the choriocapillaris (CC) layer supplies the outer retina [1, 2].

By introducing optical coherence tomography angiography (OCTA), the knowledge and evaluation of retinal capillaries were revolutionized. Studies based on OCTA are mostly based on adult populations, so we don't have any idea how age influences capillary networks in healthy children, nor do we know how normal development occurs in young children retina [3–9]. Park et al showed that a reduction in SCP, DCP, and CC vascular densities (SVD, DVD, CVD) occurs during aging [10]. For each year of increasing age, the SCP and DCP flow areas in the central 1-mm radius decreased by 0.28 mm$^2$ and 0.38 mm$^2$, respectively. This decrease was not observed in the 3-mm radius circle [3].

Also, studies on the normal adult population showed that central foveal thickness correlated with SCP and DCP [3, 11]. Furthermore, we noticed that the mild negative correlation between foveal and parafoveal VD values with age corresponds to an increased foveal avascular zone (FAZ) and decreased parafoveal flow. Yu et al. and Shahlaee et al. documented the same observation [9, 12]. Various vascular pathologies, such as neovascularization in diabetic patients may affect retinal thickness (RT) [13].

In addition, a study on children with a history of retinopathy of prematurity identified a significant positive correlation between foveal VD and foveal thickness [14]. Studies on amblyopic eyes in children showed that the amblyopic eye has a different microvasculature than fellow eye which leads to lesser RT particularly in the inner-retina [15, 16]. Although the link between RT and retinal microvasculature is already established in some pathologic processes, little is known about this association in normal childhood and developmental stages and how capillary networks impact distinct layers and zones of the macula.

The purpose of this study was to assess the correlation between RT and VD in normal healthy children and adolescents to illustrate how macular inner, middle, and outer RTs (IRT, MRT, ORT) are affected by macular perfusion.

## Method

### Participants

In this cross-sectional cohort study, healthy children aged 4 to 18 were enrolled at Farabi Eye Hospital from April 2018 to May 2020. This study was conducted under the general approval of the Institutional Review Board at Tehran University of Medical Sciences in terms of ethics. The study adhered to the tenets of the Declaration of Helsinki. This study was conducted under general approval from the Institutional Review Board of Tehran University of Medical Sciences. The consent was informed. Verbal informed consent from the participants and written consent from their parents or guardians was obtained prior to subjects' participation in the study. The study comprised healthy children and adolescents with a BCVA of 20/20, a refractive error of -1 to +1 diopter, a spherical equivalent (SE) of -0.5 to +0.5, and an intraocular pressure of less than 21 mmHg. Exclusion criteria included any condition that prevented accurate retinal imaging such as gesture and frequent blinking, and systemic, neurological, or ocular disease history, cataract, previous laser or ophthalmic surgery, amblyopia, as well as any positive past medical, surgical, and drug history.

## Imaging

A split-spectrum amplitude-decorrelation angiography algorithm (SSADA) was used on the AngioVue OCTA system version 2018,0,0,18 (Optovue RTVue XR Avanti, Optovue Inc., Freemont, California, USA). It uses an 840 nm wavelength laser to get 70,000 A-scans per second; 304 A-scans forming a B-scan. 304 vertical (Y-FAST) and horizontal (X-FAST) lines in the scanning area were used to obtain a 3D data cube and reduce the motion artifacts. Volume scans by 6×6 mm area, centered onto the fovea were performed in both eyes of each patient using 400 raster lines.

The inbuilt software defined the IRT between the internal limiting membrane (ILM) and the outer boundary of the inner plexiform layer (IPL), the MRT from the outer boundary of IPL to the outer limit of the outer plexiform layer (OPL), and the ORT from outer margin of OPL to the outer edge of the retinal pigmentary epithelium (RPE) and Bruch's membrane complex (BRM).

In this study, the RT was calculated for different sectors at the central 6 mm diameter of the macula (whole image, central at the fovea, parafovea, and perifovea) based on the Early Treatment Diabetic Retinopathy Study (ETDRS) grid. OCT determines the fovea location automatically. The foveal region was defined as a center circle with a diameter of 1.0 mm, and the parafoveal area was defined as an area of 1–3 mm width circle surrounding the foveal region and the perifoveal area of 3–6 mm circle surrounding the parafoveal area.

Automated segmentation of SCP and DCP and CC was performed using the integral software algorithm which sets the inner margin of SCP at 3 $\mu$m below the ILM of the retina and the outer boundary at 15 $\mu$m beneath the IPL, with the DCP at 15 $\mu$m beneath the IPL to 71 $\mu$m under the IPL. CC was determined as the area between 15 to 45 $\mu$m below the BM. In the binary reconstructed images, VD was calculated by the software as the relative flow density (percentage) [1]. In this study, the VD was calculated for different sectors (whole image, fovea, parafovea, perifovea, and their temporal, superior, nasal and inferior subsegmental areas) based on the ETDRS grid. Data from both eyes of all subjects were considered for analysis. The OCTA picture quality was appraised by two ophthalmologists (FG and VH). Some parts of the methods in this study are similar to those of our recent reports [17], and are described below.

## Statistical analysis

The sample size was calculated as 97 eyes with a 95% confidence interval, 10% of the change in the **unit** of VD (per percent-quantitative data), and 50% population proportion. Data analyses were performed by statistical software (SPSS software Version 21; SPSS, Inc., Chicago, IL, USA). In order to test the distribution normality of the data, the Shapiro-Wilk test was used. A partial correlation coefficient controlling for age, sex, and BMI was used to evaluate the linear correlation between VD and RT. The correlation was characterized as weakly correlated when the correlation coefficient is between 0.3 and 0.5, strongly correlated when the correlation coefficient was between 0.5 and 0.8, and very high correlation when the correlation coefficient surpassed 0.8. The Mann Whitney U test and post-hoc analysis (Dunnett's test) were performed to compare the values of nonparametric and parametric variables between groups. Considering probable correlation of measurements in two eyes of a subject, the significantly correlated variables associated with RT were entered into a multiple generalized estimating equation (GEE) analysis with the backward method to select the most proper model to test the simultaneous association of variables with the RT. In the analysis, VDs, age, sex, and BMI were considered. Also, Quasi-likelihood information criterion (QIC) was calculated to select the best fitted model. A model with the highest number of significant variables and lowest QIC was considered as a most informative model. A p-value less than 0.05 was considered statistically significant.

## Results

### Characteristics of subjects included in the analysis

A total of 108 eyes of 54 individuals were included in this study. The mean age of the subjects was 10.99 ± 0.38 years (range: 3–18 years). In total, 72% (39 cases) were males with a mean age of 10.72± 0.46 years, and 28% (15 cases) were females with a mean age of 11.47± 0.66 years. The mean weight of the group was 39.25 ± 15.37 kilograms and the mean height was 137.80 ± 21.75 centimeters.

Table 1 shows the mean values of SVD, DCP DVD and CVD at the macular area and also corresponding IRT, MRT, and ORT. Vascular density of SCP and DCP of the foveal, parafoveal, and perifoveal areas were comparable in right and left eyes (P < 0.05).

### Fovea, parafovea and perifovea

Table 2 shows results of the partial correlation coefficient between VD and RT controlling for age, sex, BMI, and characteristics of their eyes in the fovea, parafoveal, and perifoveal areas. P-values were corrected with the Bonferroni method. Foveal IRT was highly correlated with

**Table 1. OCT and OCTA characteristics of the subjects in central 6 mm diameter circle of macula.**

|  | Fovea | Parafovea | Perifovea | P value |
|---|---|---|---|---|
| SVD (%,) | 20.03 ± 7.32 | 53.13 (50.7,55.56) | 51.38 (49.695,53.065) | 0.0001 |
| Fovea |  | * | * |  |
| Parafovea | * |  | * |  |
| Perifovea | * | * |  |  |
| DVD (%,) | 37.18 (32.3,42.06) | 55.93 (52.255,59.605) | 52.46 (48.94,55.98) | 0.0001 |
| Fovea |  | * | * |  |
| Parafovea | * |  | * |  |
| Perifovea | * | * |  |  |
| CVD (%) | 75.75 (73.04,78.46) | 69.84 (66.5,73.18) | 73.47 (70.805,76.135) | 0.0001 |
| Fovea |  | * | * |  |
| Parafovea | * |  | * |  |
| Perifovea | * | * |  |  |
| IRT (µm) | 52.4 (46.95,57.85) | 108.7 (103.05,114.35) | 99.8 (94.15,105.45) | 0.0001 |
| Fovea |  | * | * |  |
| Parafovea | * |  | * |  |
| Perifovea | * | * |  |  |
| MRT (µm) | 43.4 (38.35,48.45) | 72.3 (68.65,75.95) | 58.8 (55.9,61.7) | 0.0001 |
| Fovea |  | * | * |  |
| Parafovea | * |  | * |  |
| Perifovea | * | * |  |  |
| ORT (µm) | 164.4 (156.4,172.4) | 142.7 (135.75,149.65) | 131.3 (125.4,137.2) | 0.0001 |
| Fovea |  | * | * |  |
| Parafovea | * |  | * |  |
| Perifovea | * | * |  |  |

Variables with normal distribution were presented with mean ± standard deviation and non-normally distributed data presented with median and interquartile range (Q1;Q3). P value less than 0.017 considered as significant (adjusted for multiple comparisons using the Bonferroni method).

Symbol * shows a p value less than 0.001.

Abbreviations: BMI, body mass index; CVD, choriocapillaris vascular density; DVD, deep capillary plexus vascular density; IRT, inner retinal thickness; MRT, middle retinal thickness; ORT, outer retinal thickness; SVD, superficial capillary plexus vascular density

**Table 2. This table shows partial correlation coefficients controlling for age, sex and BMI for different variables.**

| | | SVD | | | DVD | | | CVD | | |
|---|---|---|---|---|---|---|---|---|---|---|
| | | Fovea | Parafovea | Perifovea | Fovea | Parafovea | Perifovea | Fovea | Parafovea | Perifovea |
| IRT | | | | | | | | | | |
| | Fovea | 0.53 (*) | 0.43 (*) | 0.3 (*) | 0.47(*) | 0.31 (*) | 0.29 (*) | 0.36 (*) | 0.29 (*) | 0.39 (*) |
| | Parafovea | 0.18 (-) | 0.69 (*) | 0.51 (*) | 0.21 (-) | 0.49 (*) | 0.41(*) | 0.54 (*) | 0.45 (*) | 0.57 (*) |
| | Perifovea | 0.33 (*) | 0.57 (*) | 0.52 (*) | 0.30 (*) | 0.35 (*) | 0.20 (-) | 0.54 (*) | 0.36 (*) | 0.40 (*) |
| MRT | | | | | | | | | | |
| | Fovea | 0.34 (*) | -0.34 (*) | -0.18 (-) | 0.28 (*) | -0.26 (-) | -0.31 (*) | -0.23 (-) | -0.25 (-) | -0.44 (*) |
| | Parafovea | 0.17 (-) | -0.02 (-) | 0.09 (-) | 0.05 (-) | -0.1 3(-) | -0.23 (-) | 0.03 (-) | -0.01 (-) | -0.09(-) |
| | Perifovea | 0.11 (-) | -0.04 (-) | 0.13 (-) | 0.003 (-) | -0.14 (-) | -0.13 (-) | -0.04 (-) | -0.02 (-) | -0.02(-) |
| ORT | | | | | | | | | | |
| | Fovea | 0.13 (-) | 0.54 (*) | 0.34 (*) | 0.18 (-) | 0.38 (*) | 0.21 (*) | 0.49 (*) | 0.44 (*) | 0.37 (*) |
| | Parafovea | -0.03 (-) | 0.34 (*) | 0.12 (-) | -0.005 (-) | 0.21 (-) | 0.001 (-) | 0.31 (*) | 0.29(*) | 0.13 (-) |
| | Perifovea | -0.12 (-) | 0.21 (-) | -0.03 (-) | -0.04 (-) | 0.15 (-) | -0.11 (-) | 0.22 (-) | 0.26 (-) | -0.005 (-) |

P-values were corrected with the Bonferroni method and the value less than 0.005 considered as significant. Symbols in parenthesis shows P-value (- = not significant, * = P value<0.005). Abbreviations: BMI, body mass index; CVD, choriocapillaris vascular density; DVD, deep capillary plexus vascular density; IRT, inner retinal thickness; MRT, middle retinal thickness; ORT, outer retinal thickness; SVD, superficial capillary plexus vascular density

foveal SVD, DVD and CVD in all subsegments. Parafoveal IRT was highly correlated with parafoveal and perifoveal SVD and DVD and all subsegments CVD. Perifoveal IRT was highly correlated with foveal, parafoveal and perifoveal SVD, DVD and CVD except perifoveal DVD (Table 2). Foveal MRT had only a weak correlation with foveal SVD and a weak negative parafoveal SVD, and weak positive foveal and negative perifoveal DVD and a negative perifoveal CVD (Table 2). Parafoveal and perifoveal MRT was not correlated with any of the vascular layers. Foveal ORT was highly correlated with parafoveal SVD, weakly with perifoveal SVD, weakly with para and perifoveal DVD and highly correlated with CVD in all foveal, parafoveal and perifoveal areas. Parafoveal was correlated with parafoveal SVD, foveal and parafoveal CVD. Perifoveal ORT were not highly correlated with SVD, DVD, and CVD in any segment (Table 2).

## Association between RT and VD

In Fig 1, the partial correlation coefficients of SVD and RT in different ETDRS zones of the macula are shown after adjusting for age, sex, and body mass index. Nasal and inferior parafoveal and perifoveal IRT are highly correlated with SVD. Temporal parafoveal and perifoveal and superior parafoveal IRT are weakly correlated with SVD, though, perifoveal superior zone SVD was not correlated with IRT (r = 0.11, P-value = 0.2). MRT in the fovea, nasal parafoveal and perifoveal areas were negatively correlated with SVD. However, in other zones, there was no significant correlation between MRT and SVD. The ORT of fovea and temporal and inferior parafoveal areas were correlated to SVD.

In Fig 1 (middle row), the partial correlation coefficients of DVD and RT in different EDTRS zones of the macula are shown. Parafoveal nasal, inferior and temporal IRT are weakly correlated with DVD. Also, nasal and inferior perifoveal IRT were weakly correlated with DVD. However perifoveal superior zone was not correlated significantly with IRT (r = 0.03, P-value = 0.7). The MRT in the fovea and nasal parafoveal had negative even weakly correlation with the DVD. In the ORT, only the foveal thickness was very weakly correlated to CVD.

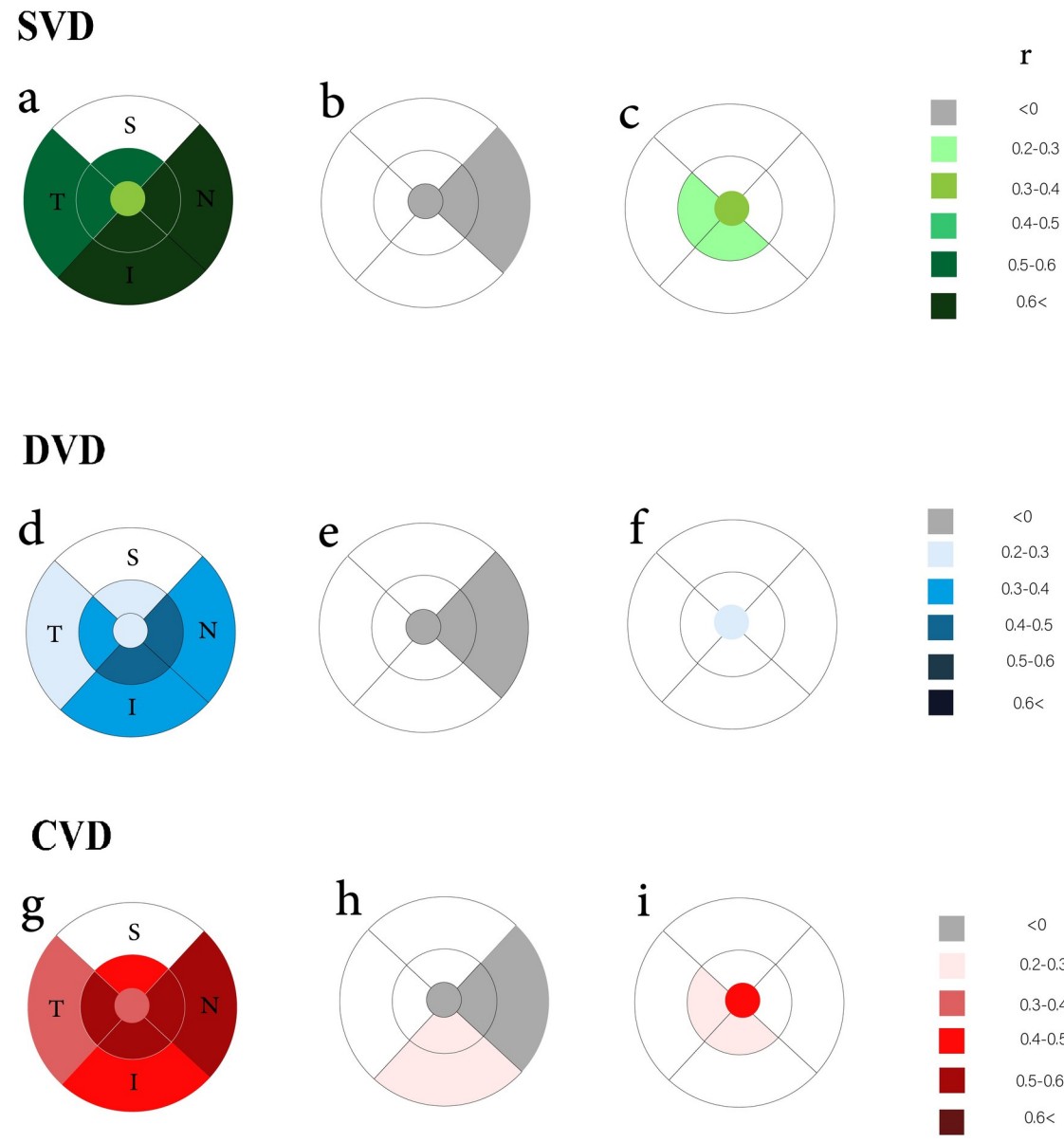

**Fig 1. These diagrams show the correlation between vascular flow density and retinal thickness in normal healthy children under 18 years old.** In each ETDRS zone magnitude of the significant correlation coefficient (r) is colored regarding the color bar. (a), (b) and (c) show the correlation between superficial capillary plexus vascular density (SVD) and inner, middle and outer retinal thicknesses, respectively. The correlation coefficient between deep capillary plexus vascular density (DVD) and inner, middle and outer retinal thicknesses depicted in images is shown in (d), (e) and (f), consequently. The correlation between choriocapillaris vascular density (CVD) and inner, middle and outer retinal thickness is shown in (g), (h) and (i), correspondingly. Abberivations: I: inferior; N: nasal; S: superior; T: temporal.

In studying the correlation of CVD and RT in different zones of the macula it is shown that with the exception of the superior perifoveal zone, all parafoveal IRT zones showed a weak to strong correlation with CVD. The nasal parafoveal and perifoveal zone were more strongly correlated with CVD. Perifoveal inferior and temporal zones showed a weak correlation with IRT. However perifoveal superior zone was not correlated significantly with IRT (r = 0.01, P-value = 0.8). MRT in the fovea, nasal parafoveal and perifoveal was negatively correlated with CVD. In parafoveal and perifoveal inferior zones, MRT correlated very weakly with CVD.

However, in other zones, there was not any significant correlation between MRT and CVD. In the ORT, the thickness of the fovea was significantly correlated to CVD.

Collectively, the nasal and inferior parafoveal and perifoveal subsegments of IRT, MRT and ORT are affected by all SVD, DVD, and CVD. Nasal parafoveal and perifoveal MRT and all three capillary layers have a constant negative association. ORT was not affected by all three layers capillaries except for CVD at fovea.

## Regression

With a GEE analysis, the variables significantly correlated with the RT were entered into models to predict the variables correlated with RT based on VDs, age, sex, and BMI. The analysis revealed that CVD was significantly associated with foveal IRT ($\beta = 0.540$, P<0.005). In addition, SVD and CVD were significantly associated with parafoveal IRT ($\beta = 1.093$, $\beta = 0.871$, respectively, P<0.005). Also, SVD, CVD, and gender could significantly affect perifoveal IRT ($\beta = 1.237$, $\beta = 0.550$ and $\beta = -5.708$, respectively, P<0.005). CVD significantly correlated with foveal MRT ($\beta = -1.145$, P<0.001). However, none of the covariables included in the model could significantly predict parafoveal or perifoveal MRT (P>0.05). The GEE analysis demonstrated that only CVD is significantly associated with foveal and parafoveal ORT ($\beta = 1.270$ and $\beta = 0.532$, respectively, P<0.001). However, none of the covariables included in the model could significantly predict parafoveal nor perifoveal ORT (P>0.05). To determine the good fit of the used models, the QIC (correlation information criterion) method was employed, and the results were evaluated using the backward method in GEE analysis (Tables 3 and S1). The result authenticated the used model in GEE analysis.

**Table 3. Association of covariables with retinal thickness.**

|  | Covariables | β | Standard error | P-value |
|---|---|---|---|---|
| **IRT** |  |  |  |  |
| Fovea |  |  |  |  |
|  | CVD | 0.54 | 0.18 | 0.003 |
| Parafovea |  |  |  |  |
|  | SVD | 1.09 | 0.38 | 0.005 |
|  | CVD | 0.87 | 0.26 | 0.001 |
| Perifovea |  |  |  |  |
|  | SVD | 1.23 | 0.26 | <0.001 |
|  | CVD | 0.55 | 0.19 | 0.005 |
|  | Gender | -5.70 | 1.87 | 0.002 |
| **MRT** |  |  |  |  |
| Fovea |  |  |  |  |
|  | CVD | -1.14 | 0.34 | 0.001 |
| **ORT** |  |  |  |  |
| Fovea |  |  |  |  |
|  | CVD | 1.27 | 0.26 | <0.001 |

Generalized estimating equation, adjusted for eyes, BMI, age, gender and vascular flow density, was used in the analyses. In para- and perifoveal middle retina and perifoveal outer retina, none of the evaluated variables were significantly associated with retinal thickness. Only factors with significant association were shown in this table. P-values were corrected with the Bonferroni method and the value less than 0.007 considered as significant. Abbreviations: BMI, body mass index; CVD, choriocapillaris vascular density; DVD, deep capillary plexus vascular density; IRT, inner retinal thickness; MRT, middle retinal thickness; ORT, outer retinal thickness; SVD, superficial capillary plexus vascular density

## Discussion

We used OCTA findings to determine the relationship between sublayer RT and retinal and CC vascular densities in a group of normal healthy children and adolescents. The study showed that the thickness in all the three layers of the retina at foveal area are significantly associated with CVD and this association is not affected by age, sex, and BMI. IRT is mainly affected by CVD and SVD at all three segments of fovea, parafovea and perifoveal area. Additionally, only perifoveal IRT was correlated with gender. Other than in the fovea, MRT and ORT are unaffected by VD of any vascular layer.

As animal-based studies showed that there is a significant neural and vascular interaction in the development of the retina [18]. Theoretically, the network of blood vessels in the retina is important for nutrient delivery and removing waste products and therefore affects RT. Also, retinal astrocytes regulate vascular sprouting during fetal ages by releasing some factors such as vascular endothelial growth factor (VEGF) [18]. Previous studies also demonstrated that age and gender affect IRT, MRT, ORT, and also retinal VD at the macular area [3, 5, 17, 19–22]. The thickness of retinal layers is altered by age in the natural developmental process. An OCT-based study on normal subjects revealed that the outer-retina and the middle-retina in adult patients are thicker than infants [23].

The results of this study highlight the fact that VD does not affect RT uniformly across ETDRS zones. It seems that RT or VD change are affected in a similar manner by age, BMI, or sex, and, consequently, RT and VD correlation is not impacted by these factors in the narrow range of age in this study. This analysis demonstrated that SVD and CVD could predict IRT. The same positive correlation between IRT and SVD has been shown in earlier studies [13–15, 24]. Oh et al. reported no correlation between the retinal structure except for ganglion cell layer-IPL thickness and the choroidal structure [24]. VD of CC was positively correlated with VD of SCP and DCP. They believed that there would be some factors that affect both GC-IPL and choroid [24]. Several studies support this finding. As their study showed that GC-IPL thickness, as well as choroid, was altered with age. Colak et al. testified that GCC was thinned in patients with migraine with aura, as with choroid [25].

Oh et al showed that there is a correlation between GCC and choroid thickness [24]. They supposed that there would be some factors that affect both GC-IPL and choroid. A study by Borrelli et al. supported this as it showed that GCC thickness decreased in dry age-related macular degeneration (AMD), a condition known to be mainly due to reduced blood flow to the CC [26].

Despite the partial correlation showing a significant positive correlation between IRT and DVD, we didn't find any statistically significant association in foveal and parafoveal zones by regression analysis. In contrary to our results, some studies reported a significant correlation between IRT and DVD [14, 15, 27]. The difference could be explained by the fact that none of the prior studies analyzed the fovea, parafovea, or perifovea separately in detail. Analysis of data also showed the correlation between SVD and IRT was present in all macular zones except for the superior perifoveal zone. In accordance, Chen et al. previously reported a significant correlation between IRT and SVD in all foveal and parafoveal areas except for the superior zone [15]. Perhaps the reason is that gravity affects mankind's retinal vascular distribution, causing no direct effect on the thickness of segments, or natural selection has protected the inferior field of vision from direct microvascular injury in the superior parafoveal and perifoveal regions of the retina.

Although partial correlation analysis showed MRT was negatively correlated with DVD, multivariate regression analysis revealed that MRT was not significantly correlated to SVD nor DVD. The only significant VD associated with MRT was CVD in the foveal zone. This could

be due to less dependence of the mid-retina to vascular perfusion or more resisttence to ische-mia in the macular EDTRS zones except for the foveal area [28]. Wu et al. performed an OCTA study on normal and high myopic subjects. They found the same negative correlation with MRT and DVD (r = -0.474, p = 0.007) [3, 27].

The same negative correlation was reported by Ye et al. in myopic participants [29]. None of the mentioned studies performed multivariate regression analysis. Same as our findings, Lavia et al. found that the DVD was not correlated with the thickness of the INL-OPL in healthy individuals [30]. This finding could be attributed to tissue levels of oxygen in different retinal layers [31]. As Lavia et al hypothesized the difference in oxygen requirement of INL and ganglion cell layer (inner retina) or ONL (outer retina) is the reason for this paradox [13]. INL is a layer with a very high density of nuclei and the cells in this layer rely on anaerobic metabolisms contrary to the GCL in the case of increased oxygen demand. This hypothesis explains why MRT is not correlated with retinal VD in most of the parafoveal and perifoveal ETDRS zones [13]. Nonetheless, in this study, we showed that there is an association between foveal MRT and CVD. Thus, it seems that CC has a role in the nutrition and function of the cells in this layer in young healthy eyes probably through facilitated direct passive oxygen and nutrient diffusion or active process through the retinal glial cells as muller cells columns. It could be hypothesized that due to its high metabolic and functional activity, the foveal area has been able to receive dual perfusion through both choroidal and retinal blood vessels through bidirectional transport route.

We identified ORT is significantly correlated with CVD in fovea and parafovea. This find-ing agrees with previous knowledge of CC function in the supply of the outer-retina [32]. All IRT, MRT, and ORT at fovea and parafovea (except for parafoveal zone MRT) but not perifo-vea were affected by CVD. The blood flows in the retina and CC are distinct from each other, but both are important to maintain homeostasis of the neural retina [24]. The CC is thought to be mainly responsible for nourishing the outer retina first through the steep gradient of oxygen tension and high blood flow in the CC, compared with retinal vasculature, that enables the choroid to deliver most of the oxygen to the retina, overcoming the barrier formed by RPE and Bruch membrane [33]. The second theory could be the action of neuroglobin acting as a transporter by removing oxygen from oxygen rich regions and delivering it to oxygen poor regions, with the potential to eliminate or alleviate hypoxia throughout the retina [34]. The dis-tribution of cytoglobin a protein that has also been anticipated to play a role in oxygen trans-port within the retina, can play a role in oxygen transport and storage in this region [35]. Theoretically, other transporters could also be involved in this vital process in a complex mul-ticompartmental tissue such as the retina.

The third possibility is the presence of clilioretinal arteries originating from posterior ciliary arteries, the common origin of choroidal vessels and CC, which can provide ancillary circula-tion to central macula and increase oxygen tension in the macular area. Prevalence of clilioret-inal artery varies from 29 to 45 percent of individuals in fluorescein angiography studies [36]. This can be supported by this finding in this study that parafoveal and perifoveal areas and mainly nasal and inferior parafoveal and perifoveal areas RT are more correlated with CVD. This study implied that the CC and retinal vasculature interact more complexly in normal chil-dren, which may affect normal development in the foveal area.

Finally, a bidirectional nutrient and oxygen transportation system may exist between retina and choroid, most likely due to the more permeable nature of RPE/Bruch mem-brane in children. The columnar unit surrounded by a Müller cell from the photoreceptor to the ganglion cell receives nutrition and oxygen in both the retinal and choroidal vascu-lar system [37, 38]. This hypothesis might be investigated in future histological and physi-ological studies.

This study demonstrated that RT is not uniformly affected by VD in different zones of macula. In the IRT, correlations of VD and RT were more prominent in inferior and nasal areas. The inferior macular zone is more likely to be affected by vascular pathologies such as Coats' disease as demonstrated in previous studies [39]. This information leads to a better understanding of the pathogenesis of this vascular disease. For instance, the presence of the abnormal vessel in the temporal zone (which is adapted to have lesser VD) interferes with the normal function and structure of the retina in macular telangiectasia type 2 in adulthood not in childhood [40]. Certainly, both functional and structural disease could alter the normal RT-VD correlation. In this study, we documented the pattern of RT-VD correlation in normal children. The correlation could be used to shed light on retinal function and physiology. Numerous diseases of the retina and choroid can be caused by improper physiological correlations. We propose further evaluation of the RT-VD correlation pattern as a valuable tool for early diagnosis and monitoring of retinal pathologies.

This study has several limitations. It is important to note that, because this is a cross-sectional study, we cannot demonstrate changes over time in VD or RT without referring to a longitudinal cohort study. A more comprehensive study should also be conducted on the development of retinal vasculature from childhood to adulthood, since its evolution may continue throughout life. Additionally, this was an OCTA-based study and segmentation error could be a limitation of the study. In addition, sensitivity parameter analysis was not performed in this analysis.

In conclusion, the retinal and choroidal VDs are correlated to RT in different segments of macula, although the correlations may differ in different zones of a layer. Further studies are needed to elucidate the causes of these alterations.

## Supporting information

**S1 Table. Values of the Quasi likelihood Information Criterion (QIC) calculated from GEE analysis of models predicting retinal thickness.**
(DOCX)

## Acknowledgments

Authors like to thank all staffs in imaging center, at Farabi Eye Hospital, Tehran, Iran, for their kind assistance during this study.

## Author Contributions

**Conceptualization:** Fariba Ghassemi.

**Data curation:** Farhad Salari, Vahid Hatami.

**Formal analysis:** Farhad Salari, Vahid Hatami, Fariba Ghassemi.

**Investigation:** Masoumeh Mohebbi, Fariba Ghassemi.

**Methodology:** Vahid Hatami, Fariba Ghassemi.

**Visualization:** Fariba Ghassemi.

**Writing – original draft:** Farhad Salari.

**Writing – review & editing:** Masoumeh Mohebbi, Fariba Ghassemi.

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
