## [Decision Letter · Decision Letter 0]

14 Apr 2022

PONE-D-22-00990Assessment of Relationship between Retinal Perfusion and Retina Thickness in Healthy Children and AdolescentsPLOS ONE

Dear Dr. Ghassemi,

Thank you for submitting your manuscript to PLOS ONE. After careful consideration, we feel that it has merit but does not fully meet PLOS ONE’s publication criteria as it currently stands. Therefore, we invite you to submit a revised version of the manuscript that addresses the points raised during the review process.

I apologize that it has taken a long time to reach this decision, but we were unable to identify a second reviewer for your submission, despite contacting many experts. So I have decided to send you the comments of the single reviewer who responded.  This reviewer made several requests with which I agree: - Please describe the sample size. Please provide more details about your statistical methods. Since you analyzed multiple variables, all p values should be corrected using *post hoc* tests for multiple comparisons. Your microperimetry results should be shown and analyzed.

We look forward to receiving your revised manuscript.

Kind regards,

Alfred S Lewin, Ph.D.

Academic Editor

PLOS ONE

Journal Requirements:

Reviewers' comments:

Reviewer's Responses to Questions

**Comments to the Author**

1. Is the manuscript technically sound, and do the data support the conclusions?

Reviewer #1: Partly

2. Has the statistical analysis been performed appropriately and rigorously? 

Reviewer #1: No

3. Have the authors made all data underlying the findings in their manuscript fully available?

Reviewer #1: Yes

4. Is the manuscript presented in an intelligible fashion and written in standard English?

Reviewer #1: Yes

5. Review Comments to the Author

Reviewer #1: I was invited to revise the paper entitled "Assessment of Relationship between Retinal Perfusion and Retina Thickness

in Healthy Children and Adolescents". It was a cross-sectional study that aimed to estimate the correlation between RT and VD in normal healthy children in order to describe the relation between RT and macula perfusionare affected by

macular perfusion. Totally were enrolled 54 healthy patients which eyes were analyzed with AngioVue OCTA System.

MAJOR OBSERVATIONS:

- Sample size estimation wastotally lacking;

- Statistical analysis section should be deeply described;

- Authors performed multiple analyses. All p-values should be corrected for multiple comparisons;

- Microperimetry retinal sensitivity should be also showed and analysed;

MINOR OBSERVATIONS:

- I suggest to show the model goodness of fit (example akaike information criterion?);

- Authors stated: "In order to test the distribution normality of the data, the Shapiro-Wilk test was used". results of these test were not reported. Some variables seems to be non-normally distributed (such as Foveal DVD or SVD). If this was confirmed, Authors should report variables as median and IQR.

6. PLOS authors have the option to publish the peer review history of their article (what does this mean?). If published, this will include your full peer review and any attached files.

Reviewer #1: No

---

## [Author Response · Author response to Decision Letter 0]

18 May 2022

PONE-D-22-00990

Assessment of Relationship between Retinal Perfusion and Retina Thickness in Healthy Children and Adolescents

PLOS ONE

Dear Prof. Alfred Lewin,

Thank you for your email. According to your comments a revised version of the manuscript (above mentioned title) that addressed the stated points is prepared.

The responses to dear editor are:

This reviewer made several requests with which I agree: - Please describe the sample size. Please provide more details about your statistical methods. Since you analyzed multiple variables, all p values should be corrected using post hoc tests for multiple comparisons. 

*All the p values were according to the below mentioned tests. 

* The sample size was calculated as 97 eyes with a 95% confidence interval.

* An analysis of variance (ANOVA) and Kruskal-Wallis tests were used for nonparametric and parametric comparisons respectively. The Mann Whitney U test and post-hoc analysis (Dunnett's test) were performed to compare the values of nonparametric and parametric variables between groups.

*We performed multiple regression analysis to investigate the most related factor to RT based on VDs, age, sex, and BMI in the eyes.

Your microperimetry results should be shown and analyzed.

**No other changes were made.

** This study didn’t receive any sort of funding. 

**The authorization of EDITORIAL MANAGER to ORCID ID was performed.

**For ethics statement the following sentence was already written in the Methods part.” Institutional review board approval was obtained from Farabi Hospital -Tehran University of Medical Sciences (Tehran-Iran) Review Board.”

Please let us know if we have to do something more.

Sincerely yours

Fariba Ghassemi MD

5. Reviewer Comments to the Author

Reviewer #1: I was invited to revise the paper entitled "Assessment of Relationship between Retinal Perfusion and Retina Thickness in Healthy Children and Adolescents". It was a cross-sectional study that aimed to estimate the correlation between RT and VD in normal healthy children in order to describe the relation between RT and macula perfusion are affected by

macular perfusion. Totally were enrolled 54 healthy patients which eyes were analyzed with AngioVue OCTA System.

MAJOR OBSERVATIONS:

- Sample size estimation was totally lacking;

- Statistical analysis section should be deeply described;

- Authors performed multiple analyses. All p-values should be corrected for multiple comparisons;

- Microperimetry retinal sensitivity should be also showed and analysed;

** The sample size was calculated as 97 eyes with a 95% confidence interval, 10% of margin of error, and 50% population proportion. 

**Thanks for the comment and the details were added to the statistic part as “An analysis of variance (ANOVA) and Kruskal-Wallis tests were used for nonparametric and parametric comparisons respectively. The Mann Whitney U test and post-hoc analysis (Dunnett's test) were performed to compare the values of nonparametric and parametric variables between groups. We performed GEE analysis to investigate the most related factor to RT based on VDs, age, sex, and BMI in the eyes.”

Unfortunately, we do not have microperimetry instrument in Farabi Hospital for comparison of retinal sensitivity. In addition, all these eyes were normal eyes with presumed normal perimetric values.

MINOR OBSERVATIONS:

- I suggest to show the model goodness of fit (example akaike information criterion?);

** Dear reviewer thanks a lot for your valuable comment. Actually, according to your comments we did the model goodness of fit compatible with GEE (Quaci likelihood criterion (QIC) and found that the previously chosen model was proper for the analysis and the results could be much reliable. I learned about that and I cordially thanks for that. The results are presented in table 3. “In the analysis, VDs, age, sex, and BMI were considered. Also, Quaci likelihood criterion (QIC) was calculated to select the best fitted model. A model with the highest number of significant variables and lowest QIC was considered as a most informative model.” 

- Authors stated: "In order to test the distribution normality of the data, the Shapiro-Wilk test was used". results of these test were not reported. Some variables seems to be non-normally distributed (such as Foveal DVD or SVD). If this was confirmed, Authors should report variables as median and IQR.

** Again, thanks a lot and the data were corrected according to the normal or non-normal distribution in Table 1.

---

## [Decision Letter · Decision Letter 1]

1 Jun 2022

PONE-D-22-00990R1Assessment of Relationship between Retinal Perfusion and Retina Thickness in Healthy Children and AdolescentsPLOS ONE

Dear Dr. Ghassemi,

Thank you for submitting your manuscript to PLOS ONE. After careful consideration, we feel that it has merit but does not fully meet PLOS ONE’s publication criteria as it currently stands. Therefore, we invite you to submit a revised version of the manuscript that addresses the points raised during the review process.

I am afraid that your description of sample size and your statistical analysis are still inadequate.

We look forward to receiving your revised manuscript.

Kind regards,

Alfred S Lewin, Ph.D.

Section Editor

PLOS ONE

Reviewers' comments:

Reviewer's Responses to Questions

**Comments to the Author**

1. If the authors have adequately addressed your comments raised in a previous round of review and you feel that this manuscript is now acceptable for publication, you may indicate that here to bypass the “Comments to the Author” section, enter your conflict of interest statement in the “Confidential to Editor” section, and submit your "Accept" recommendation.

Reviewer #1: (No Response)

2. Is the manuscript technically sound, and do the data support the conclusions?

Reviewer #1: Partly

3. Has the statistical analysis been performed appropriately and rigorously? 

Reviewer #1: No

4. Have the authors made all data underlying the findings in their manuscript fully available?

Reviewer #1: Yes

5. Is the manuscript presented in an intelligible fashion and written in standard English?

Reviewer #1: Yes

6. Review Comments to the Author

Reviewer #1: I was invited to review the revised version of the manuscript entitled "Assessment of Relationship between Retinal Perfusion and Retina Thickness in Healthy Children and Adolescents". Authors partially responded to my previous observations:

- Sample size calculation seems to be not appropriate. Authors stated "The sample size was calculated as 97 eyes with a 95% confidence interval, 10% of margin of error, and 50% population proportion". From which population? why 50% of proportion? Authors should deeply explain the calculation;

- It is unclear were Authors performed ANOVA and KWs tests;

- The lack of sensistivity parameters should be highlighted in limitation section;

- P-values correction for multiple comparisons was not performed in correlation analysis reported in table 2.

Minor observations:

- Foveal SVD in table 1 was reported as mean and SD;

- The interquartile range should be reported as (Q1;Q3);

- Table 3 should be shifted to supplementary material;

7. PLOS authors have the option to publish the peer review history of their article (what does this mean?). If published, this will include your full peer review and any attached files.

Reviewer #1: No

---

## [Author Response · Author response to Decision Letter 1]

27 Jul 2022

Dear Dr. Alfred Lewin,

Thank you for your review. We revised the last version of article entitled as “Assessment of Relationship between Retinal Perfusion and Retina Thickness in Healthy Children and Adolescents - [PONE-D-22-00990R1], addressing the points raised during the review process.

The sample size and statistical analysis were reviewed by our epidemiologist (Dr Siamak Sabour) and the needed corrections were done accordingly. If there is anything else we can do, please let us know. 

Sincerely yours,

 Fariba Ghassemi MD

PONE-D-22-00990R1

Reviewer #1: I was invited to review the revised version of the manuscript entitled "Assessment of Relationship between Retinal Perfusion and Retina Thickness in Healthy Children and Adolescents". Authors partially responded to my previous observations:

- Sample size calculation seems to be not appropriate. Authors stated "The sample size was calculated as 97 eyes with a 95% confidence interval, 10% of margin of error, and 50% population proportion". From which population? why 50% of proportion? Authors should deeply explain the calculation;

** Dear Reviewer

Thanks for your valuable comments. We learned a lot from these comments. For your information, we consulted with our epidemiologist (Dr Siamak Sabour) and will respond according to his opinion. In the continuity of our work on normal variations of vascular density of adult eyes, we were asked to do the same study in the eyes of children. The first purpose of the study was to evaluate the vascular density of the eyes that was a quantitative type of data which were reported as percentage. 

The above mentioned 10% of the change in the vascular density would be considered as significant. It was better to be said 10% of the change in the unit (per percent). The “Power and Sample Size Calculation Software version 3.1.2” was used for evaluating the appropriateness of sample size as another software. You can see the results below.

We are planning a study of a continuous response variable from independent control and experimental subjects with 1 control(s) per experimental subject. In a previous study the response within each subject group was normally distributed with standard deviation of 10. If the true difference in the experimental and control means is 10 units, we will need to study 17 eyes in each age categories to be able to reject the null hypothesis that the population means of the experimental and control groups are equal with probability (power) 0.80. The type I error probability associated with this test of this null hypothesis is 0.05.

We think that from our sentence in the rebuttal letter as “10% of margin of error, and 50% population proportion", you have considered it as a dichotomous outcome analysis with the results shown below. But our analysis was a descriptive and analytic test on quantitative parameters only. In the case of using such dichotomous outcome analysis, we needed to have a sample size by 376. 

Although, we lost some cases due to the limitation of CORONA crisis in the course of the study, according to the used sample size within each age group (by nearly 24 cases in each 4 age groups of studied children and adolescents ranged from 3 to 18 YO), the power was calculated to be 92%. 

We were planning a study of subjects in which we will regress their values of yvar against xvar. Prior data indicate that the standard deviation of xvar is 0.1 and the standard deviation of the regression errors will be 0.2. If the true slope of the line obtained by regressing yvar against xvar is 0.5, we will need to study 10 subjects to be able to reject the null hypothesis that this slope equals zero with probability (power) 0.0.80. The Type I error probability associated with this test of this null hypothesis is 0.05.

- It is unclear were Authors performed ANOVA and KWs tests; 

**Thanks a lot for meticulous attention. We used ANOVA and KW tests for evaluating our four age groups of children, because our parameters were quantitative parameters in our four aging groups as that reported from our previous study on these children and adolescents.

“Ghassemi F, Hatami V, Salari F, Bazvand F, Shamouli H, Mohebbi M, Sabour S. Quantification of macular perfusion in healthy children using optical coherence tomography angiography. Int J Retina Vitreous. 2021 Oct 2;7(1):56. doi: 10.1186/s40942-021-00328-2. PMID: 34600586; PMCID: PMC8487563.”

This study is a second look for the data, evaluating the correlation between the thickness and the vascular density in macular area.

- The lack of sensitivity parameters should be highlighted in limitation section.

**Thanks for the comment. This is added as “In addition, sensitivity parameter analysis was not performed in this analysis” to the limitation section. 

- P-values correction for multiple comparisons was not performed in correlation analysis reported in table 2.

** The correction was done according the Bonferroni method and the needed change was performed in the result part and highlighted. 

Minor observations:

- Foveal SVD in table 1 was reported as mean and SD;

**As stated in the legend of the table, the variables with normal distribution are presented with mean ± standard deviation and non-normally distributed data presented with median (interquartile range or min - max). 

- The interquartile range should be reported as (Q1; Q3);

**Thanks, it is corrected.

- Table 3 should be shifted to supplementary material;

**Thanks, Table 3 shifted to supplementary material.

---

## [Decision Letter · Decision Letter 2]

1 Aug 2022

Assessment of Relationship between Retinal Perfusion and Retina Thickness in Healthy Children and Adolescents

PONE-D-22-00990R2

Dear Dr. Ghassemi,

We’re pleased to inform you that your manuscript has been judged scientifically suitable for publication and will be formally accepted for publication once it meets all outstanding technical requirements.

Kind regards,

Alfred S Lewin, Ph.D.

Section Editor

PLOS ONE

Additional Editor Comments (optional):

Reviewers' comments:

Reviewer's Responses to Questions

**Comments to the Author**

1. If the authors have adequately addressed your comments raised in a previous round of review and you feel that this manuscript is now acceptable for publication, you may indicate that here to bypass the “Comments to the Author” section, enter your conflict of interest statement in the “Confidential to Editor” section, and submit your "Accept" recommendation.

Reviewer #1: All comments have been addressed

2. Is the manuscript technically sound, and do the data support the conclusions?

Reviewer #1: Yes

3. Has the statistical analysis been performed appropriately and rigorously? 

Reviewer #1: Yes

4. Have the authors made all data underlying the findings in their manuscript fully available?

Reviewer #1: No

5. Is the manuscript presented in an intelligible fashion and written in standard English?

Reviewer #1: Yes

6. Review Comments to the Author

Reviewer #1: Authors properly addressed all comments raised. The statistical analysis was improved and now it was described more in depth. Now, the paper can be accepted for publication.

7. PLOS authors have the option to publish the peer review history of their article (what does this mean?). If published, this will include your full peer review and any attached files.

Reviewer #1: **Yes: **Giuseppe Di Martino

---

## [Editor Report · Acceptance letter]

3 Aug 2022

PONE-D-22-00990R2 

Assessment of Relationship between Retinal Perfusion and Retina Thickness in Healthy Children and Adolescents 

Dear Dr. Ghassemi:

I'm pleased to inform you that your manuscript has been deemed suitable for publication in PLOS ONE. Congratulations! Your manuscript is now with our production department. 

Kind regards, 

on behalf of

Dr. Alfred S Lewin 

Section Editor

PLOS ONE